# Salinity Induced Antioxidant Defense in Roots of Industrial Hemp (IH: *Cannabis sativa* L.) for Fiber during Seed Germination

**DOI:** 10.3390/antiox11020244

**Published:** 2022-01-27

**Authors:** Naveen Dixit

**Affiliations:** Department of Agriculture Food and Resources Sciences, University of Maryland Eastern Shore, Princess Anne, MD 21853, USA; fnaveenkumar@umes.edu

**Keywords:** Hemp, oxidative stress, roots, antioxidant defense, salinity, ascorbate-glutathione cycle

## Abstract

Global climate change induced sea level rise, rainfed agriculture, poor quality irrigation water, and seawater intrusion through interconnected ditches and inland waterways cause soil salinity in inland and coastal areas. To reclaim and prevent further soil erosion, salt tolerant crops are required. Industrial Hemp (IH: *Cannabis sativa* L.) is used for food, fiber, and medicinal purposes throughout the world. In spite of that, little is known about the salt tolerance mechanisms in IH. Seed germination and development of the roots are the primary events in the life cycle of a plant, which directly interact with soil salinity. Therefore, in vitro germination experiments were conducted on the roots of 5-day-old seedlings using four varieties (V1: CFX-2, V2: Joey, V3: Bialobrzeskie, and V4: Henola) of IH for fiber. Five salinity treatments (0, 50, 80, 100, 150, and 200 mM NaCl) were used to screen the IH varieties on the basis of I: seed germination percent (SGP), II: quantitative morphological observations (root length (RL) and root fresh weight (RFW)), III: oxidative stress indices (hydrogen peroxide (H_2_O_2_) and lipid peroxidation), and IV: antioxidant defense system comprises of superoxide dismutase (SOD), catalase (CAT), guaiacol peroxidase (GPOD), ascorbate peroxidase (APOD), glutathione reductase (GR). The varieties V1 and V3 showed salt tolerance up to 100 mM by maintaining higher SGP, less reduction in RL and RFW. These roots in V1 and V3 showed lower levels of H_2_O_2_ and lipid peroxidation by displaying higher activities of SOD, CAT, GPOD, APOD, and GR while a reciprocal trend was observed in V4. However, roots in V2 showed higher activities of antioxidant enzymes with lower levels of H_2_O_2_ and lipid peroxidation, but showed declines in RL and RFW at 80 mM NaCl onward. Roots in V4 were the most susceptible to NaCl stress at 50 mM and onward.

## 1. Introduction

Soil salinity is a global ecological issue and causes soil degradation by modifying its properties, structure, and function [1,2]. Both natural and anthropogenic factors are responsible for the continuous increase in salt affected soils comprised of climate change, sea level rise, seawater intrusion, flash floods, alternating wet and dry periods, persistent drought, poor water quality, and poor irrigation practices [2,3]. IH for fiber is an economically important crop and grown worldwide for cellulosic and woody fibers [4]. IH fiber demand is growing for the textile and manufacturing industry [5]. Fiber IH generates enormous quantities of biomass within a short time span and also improves soil structure [4]. In addition, IH for fiber can be used to reclaim soils affected by the monoculture of cereals and legumes [6]. Hemp is adapted to a wide range of abiotic stresses including soil salinity, but large genetic variability exists among the varieties for salt tolerance [5,7,8]. It is therefore imperative to explore salt tolerance and associated mechanisms in existing IH germplasm to provide timely information to growers.

Soil salinity causes salt build up in the root zone and adversely affects all the stages of plant life cycle through osmotic and ionic stresses [9]. Both of these stresses also cause production of reactive oxygen species (ROS: superoxide radicals (O^•−^), hydrogen peroxide (H_2_O_2_), hydroxyl radical (^•^OH), and singlet oxygen (^1^O_2_) in multiple cell organelles [10,11].

In plants, seed germination and root establishment are crucial events for successful stand establishment [3,7,12]. However, salinity inhibits both the processes and jeopardizes crop growth and yield [9]. Seed germination requires absorption of water to initiate metabolic events for the emergence of the seed radicle [13]. Higher salt concentration in soil solution creates physiological drought and impedes water movement within the seed, thus inhibiting seed germination [13,14]. IH salt sensitive variety, Bamahuoma showed poor and delayed seed germination at 100–300 mM NaCl in comparison to salt tolerant variety YM5 [7]. Several mechanisms have been proposed for salt tolerance during seed germination including type of seed structure, seed coat anatomy, and seed size/food reserve [7,15].

Salinity induced root inhibition is evident in IH and other crop plants [3,12]. Root growth is an outcome of cell division and expansion, in saline environments, high osmotic potential, ion toxicity, and toxic levels of H_2_O_2_ severely impaired root growth and development [10,16,17]. Root length declined in both YM5 and Bamahuoma IH varieties with a successive increase in NaCl and Na_2_CO_3_ concentrations [12]. Salt (50–150 mM NaCl) induced H_2_O_2_ production in *Oryza sativa* L. roots caused significant decline in RL and RFW [18]. Similarly, exogenous application of H_2_O_2_ (0.1–9 mM) in *Triticum aestivum* L. em Thell roots decreased hydraulic conductivity within 30 min after exposure, which might further contribute to a decline in RFW and RL by limiting water availability [17]. Reduction in root growth in salt regimes seems to be an adaptive response for plant survival at the expense of plant size under restricted water and nutrient supply [19]. RL is inhibited by higher GPOD activity in *T. aestivum* L. em. Thell roots following NaCl (150 mM) treatment [17]. GPOD mediated thickening/lignification of cell walls prevents root cell elongation and thus, maintains turgor in the salt stressed roots [17,19]. Higher GPOD activity is associated with salt tolerance in roots [11,20].

Roots with higher levels of antioxidant enzymes successfully manage salinity induced ROS and can be used as an indicator for salt tolerance in IH [10,11]. Multiple pathways operate in roots to generate H_2_O_2_ from O^•−^ by the activity of superoxide dismutases (SODs), GPOD, and polyamine oxidase [21,22,23]. There are reports that have shown salinity induced generation of O^•−^ by membrane bound NAPDH oxidase and mitochondrial electron transport chain in root cells [24,25]. SODs are regarded as a protecting enzyme against O^•−^. SOD converts O^•−^ in less toxic H_2_O_2_, but higher concentration of H_2_O_2_ is detrimental for root survival [10,26]. Higher levels of H_2_O_2_ in association with free Cu^+^ or Fe^2+^ produce highly reactive ^•^OH, which promotes lipid peroxidation [21]. Lipid peroxidation is a robust marker of oxidative damage in cells [27]. Salt induced increased lipid peroxidation has been observed in roots of *Lycopersicon esculentum* L. and *O. sativa* L. [11,28]. The increase was more in salt sensitive varieties in comparison to salt tolerant [11,28]. In addition to non-enzymatic lipid peroxidation, lipoxygenase mediated lipid peroxidation is also reported in saline conditions [29,30]. Recently, lipoxygenase induced ^1^O_2_ production was reported in osmotically stressed roots of *Arabidopsis thaliana* L. and suggested to be associated with lipid peroxidation and root growth inhibition [31].

Salt tolerant crops maintain root plasticity at morphological and anatomical levels [32]. Up-regulation of H_2_O_2_ metabolizing enzyme machinery can be a part of root metabolic plasticity against the salinity induced oxidative stress in IH. H_2_O_2_ formation and degradation is tightly regulated in cells as H_2_O_2_ also acts as a signaling molecule during abiotic and biotic stresses [33]. Combined action of housekeeping antioxidant enzymes CAT, GPOD, and ascorbate peroxidase (APOD) keeps the steady state levels of H_2_O_2_ in normal conditions [33]. CAT and APOD both detoxify H_2_O_2_, but CAT has low affinity (*K_m_* = 20–124 mM range) for H_2_O_2_ and removes bulk of the H_2_O_2_ from the system while APOD has high affinity (*K_m_* = 20–74 µM range) and thus fine tunes the H_2_O_2_ level [10,33,34,35]. Peroxisomes and mitochondria isolated from NaCl (100 mM) treated roots of salt tolerant *L. pennellii* (closely related wild species of tomato) showed higher activities of SOD, CAT, and APOD with significant decline in H_2_O_2_ and MDA levels (11). Similarly, GPOD activity was also higher in these cell organelles and contributed to salt tolerance. GPOD also detoxifies H_2_O_2_ by using various reductants [36]. Higher GPOD activities (1.5-fold) were detected in salt tolerant Pistachio (*Pistacia vera* L.) rootstock UCB-1 at 16 dS/m in comparison to controls, while the lowest values were evident in salt sensitive cultivars [37].

APOD works in cooperation with glutathione reductase (GR) though Ascorbate–Glutathione cycle or Foyer-Halliwell-Asada pathway [38]. However, this redox-based cycle requires NAPDH to convert H_2_O_2_ into water [38]. GR generates reduced glutathione (GSH), which ultimately produces reduced ascorbate, required for APOD activity through an intermediate enzyme dehydroascorbate reductase [37]. APOD and GR activity increased in NaCl (150 mM) treated roots of *O. sativa* L. with a corresponding increase in *OsAPX* and *OsGR* gene expression [39]. Similarly, higher GR activity was reported in roots of *Pistacia vera* L. rootstock UCB-1 at 16 dS/m, which showed GR mediated protection against higher levels of H_2_O_2_ in salt enriched environments [37]. Little information is available on oxidative stress management in roots of IH for fiber [8,40,41]. However, most of this information is related to heavy metal toxicity and associated root responses [8,40,41]. Evaluation of the antioxidant defense system in roots of IH germplasm will provide crucial information on how IH roots respond to salinity and deploy protective mechanisms for survival and crop establishment. In the current work, root antioxidant defense system-based screening has been performed using saline conditions for commercially available IH germplasm for fiber.

## 2. Materials and Methods

### 2.1. Plant Material and Culture

Seeds of four (V1: CFX-2, V2: Joey, V3: Bialobrzeskie, and V4: Henola) commercially available varieties of IH for fiber were purchased from Kings Agriseeds Inc., Lancaster, PA, USA. Seeds were tested to find out the SGP in control conditions prior to experimentation and results confirmed >90% seed germination in all the varieties as provided on the label by the company. There is no difference in the 1000 seed weight (V1: 21.5 g, V2: 21.4 g, V3: 19.7 g, and V4: 20.5 g) among the tested varieties. Uniform sized and intact seeds were selected for germination and seedling root studies. Seeds were sterilized with 2.5% sodium hypochlorite for 10 min followed by extensive washing with distilled water and air drying [42]. Sterilized seeds were transferred to a 14 cm diameter Petri dish containing two layers of filter paper (P5 grade, Fisher Scientific, Hampton, PA, USA). Seeds were placed randomly between layers of filter paper and soaked with 11 mL of five different NaCl concentrations (0, 50, 80, 100, 150, and 200 mM). Petri dishes were kept at 27 ± 2 °C for 5 days using a G24 environmental incubator (New Brunswick Scientific Co., Inc., Edison, NJ, USA). Seeds were considered germinated upon display of 1 mm long radicle [7]. Seed germination percent was calculated by dividing the number of seeds germinated by the total number of seeds and multiplying by 100. RL was measured at day 5 by using a ruler and length was expressed in cm. Same root samples were used for the measurement of RFW and the weight expressed in mg. Root samples were collected at day 5 and processed for enzymes, H_2_O_2_, and lipid peroxidation analysis. Three independent experiments were conducted during June–August 2021. Fifteen replicates were used for the estimation of all the biochemical parameters and forty-five replicates for morphological parameters (SGP, RL, and RFW). Morphological parameters were evaluated up to 200 mM NaCl concentration while biochemical assays were evaluated only up to 150 mM NaCl.

### 2.2. Assay of H_2_O_2_ Generation

Root tissue samples (0.25 g) of IH were grounded using mortar and pestle in 0.1% trichloroacetic acid and centrifuged (Centrifuge 5430 R, Eppendorf, Enfield, CT, USA) at 4 °C for 20 min at 14,000× *g*. The supernatant (0.3 mL) was collected and thoroughly mixed with 1.7 mL of potassium phosphate buffer (pH 7.0) and 1 mL of 1 M potassium iodide (KI) solution. The mixture was incubated for 5 min and optical density was measured 390 nm using the spectrophotometer (UV-1900, Shimadzu, Kyoto, Japan). A standard curve of H_2_O_2_ was prepared from the known concentrations of H_2_O_2_ to calculate the unknown concentrations of H_2_O_2_ in root samples. The concentration of H_2_O_2_ was expressed as nmole g^−1^ Fwt. [43].

### 2.3. Determination of Lipid Peroxidation

Lipid peroxidation was determined by using the method of Heath and Packer (1968) with modifications and measured as malonic dialdehyde (MDA) content in 0.25 g roots tissue sample [44,45]. The concentration of MDA was calculated by subtracting the absorption at 532 nm with non-specific absorption at 600 nm by using the extinction coefficient of 155 mM^−1^ cm^−1^. The MDA concentration was expressed as nmole g^−1^ Fwt.

### 2.4. Root Preparation for Enzyme Assay

IH root tissue samples (0.25 g) were grounded using a pre-chilled pestle and mortar in liquid nitrogen and then homogenized in an ice cold potassium phosphate (pH 7.0) buffer containing ascorbate (5 mM), polyvinylpyrrolidone (2%), ethylenediaminetetraacetic acid (EDTA; 0.5 mM), and phenylmethylsulphonyl fluoride (0.5 mM).

The enzymes extraction procedure was carried out at 4 °C. The homogenate was centrifuged at 14,000× *g* for 30 min. To maintain the electrophoretic mobility of CAT, 10 mM dithiothreitol (DTT) was added to each aliquot samples for CAT assays [46]. All the supernatants were processed by using Sephadex G-25 (PD-10 column, Pharmacia, Jersey City, NJ, USA) pre-equilibrated with 50 mM potassium phosphate buffer (pH 7.0). Enzyme assays were conducted in a final volume of 3 mL. The following volumes (25–30 μL for SOD; 80–100 μL for CAT; 50–100 μL for GPOD; 40–100 μL APOD; 50–100 μL for GR) of aliquots were used for enzymatic assays.

SOD activity was measured spectrophotometrically using the method of Dhindsa et al. [47]. Preliminary work was conducted by overnight dialysis (cellulose ester dialysis tubing; MWCO 8000–10,000 Dalton) of crude extract to observe the interference by low molecular weight impurities. No interference was observed in these preliminary studies [48]. All the assays were conducted in a final volume of 3 mL containing 50 mM potassium phosphate buffer (pH 7.8), 1.72 mM nitroblue tetrazolium, 201 mM methionine, enzyme extract (20–30 µL), and riboflavin (0.12 mM). The reaction was initiated by keeping the tubes under the two 75 W fluorescent bulbs for 10 min. Absorbance of the colored product was measured at 560 nm. One unit of SOD activity was defined as the amount of enzyme required for the 50% inhibition of the rate of reaction in the absence of the enzyme [47]. SOD specific activity was expressed as units min^−1^ mg^−1^ protein.

CAT specific activity was determined by the rate of decomposition of H_2_O_2_ at 240 nm by using extinction coefficient = 39.4 mM cm^−1^ [49]. The reaction mixture contained 10 mM H_2_O_2_, enzyme extract (50–100 μL), and 50 mM potassium phosphate buffer (pH 7.0). CAT specific activity was expressed as µmole min^−1^ mg^−1^ protein.

GPOD activity was determined by the increase in absorption at 470 nm due to oxidation of guaiacol by using extinction coefficient = 26.6 mM cm^−1^ [49]. The reaction mixture contained 10 mM H_2_O_2_, 0.05% guaiacol, enzyme extract (40–100 μL), and 50 mM potassium phosphate buffer (pH 7.0). GPOD specific activity was expressed as µmole min^−1^ mg^−1^ protein.

APOD activity was determined by following the rate of ascorbate oxidation at 290 nm using extinction coefficient = 2.8 mM cm^−1^ [50]. The reaction mixture contained 1.0 mM H_2_O_2_, 0.1 mM EDTA, 0.25 mM ascorbic acid, enzyme extract (50–100 μL), and 50 mM potassium phosphate buffer (pH 7.0). APOD specific activity was expressed as µmole of ascorbate oxidized min^−1^ mg^−1^ protein.

GR activity was determined by the increase in absorbance at 412 nm over a period of 5 min at 25 °C [51]. The reaction mixture contained 3 mM 5,5-dithio-bis-2-nitrobenzoic acid, 2 mM NADPH, 0.1 mM EDTA, 0.67 mM oxidized glutathione (GSSG), enzyme extract (50–100 μL), and 50 mM potassium phosphate buffer (pH 7.5). GR specific activity was expressed as nmole min^−1^ mg^−1^ protein. Protein concentration was measured by the Bradford method [52].

### 2.5. Statistical Analysis

Three experiments were conducted as a completely randomized design and fifteen replicates with 10 seeds each were used for each treatment. All biochemical assays were repeated three times with five replicates on each seedling (*n* = 15) while forty-five replicates were used for morphological parameters (SGP, RL, and RFW). A two-way ANOVA was used to find differences between varieties and salt concentrations and their interactions. The means per plant were calculated and analysis of variance (SAS OnDemand for Academics) was carried out to separate these means by a protected least significant difference (LSD) at *p* < 0.05. The standard error (SE) of the mean was calculated.

## 3. Results

### 3.1. Seed Germination Percent

Seed germination rate was non-significant among the four varieties in control conditions and varied between 93 to 90% (V1: 92%, V2: 90%, V3: 93%, V4: 91%; Figure 1A). The SGP declined in V1 and V3 at 150 mM NaCl onward and the decline was 33% and 31%, in V1 andV3, respectively, at 200 mM NaCl in comparison to control. However, SGP declined in V2 and V4 at 80 mM NaCl onward and this decline was 67% and 70% in V2 and V4, respectively, at 200 mM NaCl in comparison to control. The highest decline in SGP was observed in V4. The varieties V1 and V3 showed non-significant differences in SGP up to 100 mM salt concentration. However, significant differences were observed in SGP in V4 at 80 mM to 200 mM salt concentration. The decline in SGP was 11%, 28%, 57%, and 70% at 80, 100, 150, and 200 mM NaCl concentrations, respectively, in comparison to control. The variety V1 and V3 showed increased SGP at 80 mM to 200 mM NaCl in comparison to V2 and V4

### 3.2. Root Length

The longest RL was observed in V3 followed by V2 > V1 > V4 in control seedlings (Figure 1B). Root length declined in all the varieties with successive levels of NaCl concentration. The decline was 78%, 83%, 74%, and 89%, respectively, in V1, V2, V3, and V4 at 150 mM NaCl. The highest decline (98–99%) in RL was observed at 200 mM NaCl in all the varieties. The minimum decline in RL was observed in V1 and V3 at 50 mM to 100 mM NaCl. The decline was 26% and 35% at 100 mM NaCl in V1 and V3, respectively. However, this decline was more severe in V4 (71%) and V2 (41%).

### 3.3. Root Fresh Weight

The varietal differences were observed in RFW (Figure 1C). The maximum RFW was recoded in V3 followed by V2 > V1 > V4. Root fresh weight declined with increasing levels of NaCl concentrations in all the tested varieties. This decline was 72%, 72%, 69%, and 91% in V1, V2, V3, and V4, respectively, in comparison to controls at 150 mM NaCl. The minimum decline in RFW was observed in V1 (1.3 times) and V3 (1.3 times) at 100 mM NaCl, while it was the highest in V4 (2.6 times) and V2 (1.6 times) in comparisons to controls.

### 3.4. Hydrogen Peroxide

The levels of H_2_O_2_ were non-significant among the varieties in control seedlings and varied from 0.27 to 0.29 nmole g^−1^ Fwt. (Figure 2A). These levels declined in V1 (1.8 times), V2 (1.2 times), and V3 (1.8 times) up to 100 mM NaCl in comparison to controls, but showed a consistent increase in V4 (two times). However, all the varieties showed increased H_2_O_2_ levels at 150 mM NaCl. The highest H_2_O_2_ levels were observed in V4. These H_2_O_2_ levels remained higher in V4 in comparison to other varieties at all the tested NaCl concentrations.

### 3.5. Lipid Peroxidation

The MDA content was similar in all the four varieties and varied from 4.16 to 4.19 nmole g^−1^ Fwt. in control roots (Figure 2B). The rate of lipid peroxidation decreased in V1 (1.3 times), V2 (1.1 times), and V3 (1.3 times) up to 100 mM NaCl and then increased at 150 mM. The highest levels of lipid peroxidation were observed in V4 at all levels of NaCl concentration. The rate of lipid peroxidation was 1.0, 1.1, 1.5, and 1.8 times at 50, 80, 100, and 150 mM NaCl concentrations, respectively, in comparison to controls.

### 3.6. Superoxide Dismutase

The SOD specific activity increased in all the varieties up to 100 mM NaCl and, thereafter, declined at 150 mM (Figure 2C). The highest SOD specific activity was observed in V3, followed by V1 > V2 > V4 at all levels of NaCl concentrations. The activity was 70%, 78%, 100%, and 51% higher in V1, V2, V3, and V4 at 100 mM NaCl in comparison to control roots.

### 3.7. Catalase

The CAT specific activity was significantly different in tested varieties in control conditions (Figure 3A). The maximum CAT specific activity (0.07 µmole min^−1^ mg^−1^ protein) was observed in V4 in control regimes. CAT specific activity increased in V1 (52%), V2 (39%), and V3 (43%) up to 100 mM NaCl and then declined at 150 mM. However, CAT specific activity slightly increased (3%) at 50 mM in V4, and then subsequently declined by 85% at 150 mM NaCl.

### 3.8. Guaiacol Peroxidase

The GPOD specific activity showed a consistent increase in all the varieties at all the levels of NaCl concentration (Figure 3B). The highest GPOD specific activity (0.43 µmole min^−1^ mg^−1^ protein; 100 mM) was observed in V3 followed by V1 > V2 > V4 at 100 and 150 mM NaCl. The lowest increase in GPOD specific activity was observed in V4 at all the levels of NaCl concentration in comparison to other varieties. The increase in GPOD specific activity was only 3 times in V4 at 150 mM NaCl in comparison to 11 times, 7.4 times, and 12 times in V1, V2, and V3, respectively.

### 3.9. Ascorbate Peroxidase

The APOD specific activity showed significant differences among varieties in control regimes (Figure 3C). The highest APOD specific activity (0.01 µmole ascorbate oxidized min^−1^ mg^−1^ protein) was observed in V4 followed by V2 > V3 > V1. APOD specific activity increased in roots with an increase in NaCl concentration up to 100 mM and then declined at 150 mM in V1, V2, and V3. However, APOD specific activity increased up to 80 mM NaCl in V4 and thereafter declined. This decline was 22% and 67% at 100 and 150 mM NaCl in comparison to controls. The maximum APOD specific activity was observed in V3 (0.24 µmole ascorbate oxidized min^−1^ mg^−1^ protein) followed by V2 (0.22 µmole ascorbate oxidized min^−1^ mg^−1^ protein) > V1 (0.21 µmole ascorbate oxidized min^−1^ mg^−1^ protein) > V4 (0.08 µmole ascorbate oxidized min^−1^ mg^−1^ protein) at 100 mM NaCl.

### 3.10. Glutathione Reductase

The varietal differences in GR specific activity were apparent in control conditions (Figure 4). GR specific activity was the highest (4.0 nmole min^−1^ mg^−1^ protein) in V3 followed by > V4 > V1 > V2 control roots. GR specific activity consistently increased in V3 and V1 up to 150 mM NaCl, while it declined in V2 at 150 mM. The GR specific activity only increased up to 50 mM NaCl in V4 and thereafter declined at successive levels of NaCl concentration. The specific activity of GR was the lowest in V4 at all NaCl treatments in comparison to other tested varieties.

## 4. Discussion

Soil salinity is one of the major crop production constraints in the changing climatic conditions [2]. Higher salt levels in soil reduce soil productivity, stability, structure, and biodiversity [2]. Almost one-third of irrigated land in the world is contaminated with salt by rising sea levels, saltwater intrusion, groundwater irrigation, and poor drainage and irrigation systems [53,54]. Salinity imparts adverse effects on each and every stage of plant growth and development [55]. Seed germination is an initial event in the life cycle of a plant and experiences direct interaction with salt in soil solution [7]. Successful stand establishment and ground coverage is dependent on seed germination [56]. Among the tested IH varieties, SGP declined in V1 (3%) and V3 (5%) at 150 mM NaCl while at 80 mM in V2 (3%) and V4 (12%) in comparison to controls (Figure 1A). These differences were higher (V1: 33%; V2: 67%; V3: 31%; V4: 70%) at 200 mM NaCl in comparison to controls. Salt tolerance in terms of SGP was the highest in V1 followed by V3 > V2 > V4.

Salt enriched soils can generate higher osmotic and lower water potential in the germinating medium and thus create physiological drought and inhibit seed germination [7,57]. Moreover, ion toxicity can further suppress seed germination [7]. Most of the glycophytes showed growth inhibition and plant death at 100–200 mM NaCl concentration [58]. However, genetic variability exists for salt tolerance in IH and depends on type of species and variety [7]. Recent work on IH showed that cultivar Yunma 5 (fiber use) had higher salt tolerance in comparison to Bamahuoma (seed use) for seed germination at 150 mM NaCl, Na_2_SO_4_, and NaCO_3_, and 250 mM NaHCO_3_ concentration [7]. Similarly, in our work V1 (89%) and V3 (88%) showed higher salt tolerance up to 150 mM NaCl while V4 (34%) appeared to be a salt sensitive variety in vitro germination study. Several mechanisms are proposed for salt tolerance during seed germination including food reserve mobilization and seed anatomy, which can provide sufficient energy and prevent salt intrusion in seeds prior to radical emergence [7,59].

The higher SGP at 100 mM NaCl in V1 (26%) and V4 (35%) accompanied by minimum decline in RL in comparison to V4 (71%) and V2 (41%; Figure 1B). A very similar trend was observed in RFW (Figure 1C). The minimum decline in RFW was observed in V1 and V3 (1.3 times) and the maximum in V4 (2.6 times) and V2 (1.6 times) at 100 mM NaCl. The smaller decline in RL and RFW showed better water status in V1 and V3 roots in comparison to V4 and V2 at 100 mM NaCl. Roots are the primary organ in plants, which directly interact with salts in soil [10]. Root growth and development is a polarized mechanism and is based on cell division and elongation [10]. Salinity interferes with both the processes though osmotic and ionic effects [10,16]. Ten-days-old *Arabidopsis thaliana* L. seedling roots in 0.5% NaCl showed a decrease in cell production and mature cell length. This decrease in cell number is mainly due to a smaller number of dividing cells in the meristematic zone of the root [16]. In addition, there is evidence, which showed salinity induced H_2_O_2_ formation in roots with concomitant decline in root hydraulic conductivity in *T. aestivum* L. em. Thell [17].

Our data showed an increase in root H_2_O_2_ levels with an increase in NaCl concentration in V4 from 0.27 (control) to 0.54 nmole g^−1^ Fwt. at 100 mM NaCl (Figure 2A). However, levels of H_2_O_2_ declined in V1 (100 mM: 0.16 nmole g^−1^ Fwt.), V2 (100 mM: 0.22 nmole g^−1^ Fwt.), and V3 (100 mM: 0.16 nmole g^−1^ Fwt.) at subsequent concentrations of NaCl except 200 mM. In *T. aestivum* L. em. Thell, root hydraulic conductivity declined by 2 to 3 times with 0.1–9 mM H_2_O_2_ exposure [17]. Higher H_2_O_2_ levels in V4 roots are responsible for the decline in RL and RFW at 50 mM NaCl onward. The smaller decline in RL and RFW of V1 and V3 is due to better management of H_2_O_2_ levels (Figure 2A). Salinity in association with osmotic and ionic effects can generate oxidative stress [60]. It is apparent that V1, V3, and V2 can manage H_2_O_2_ levels up to 100 mM NaCl, which failed in V4 and reflected on maximum reduction in RL (71%) and RFW (2.6-times) in comparison to controls. In V2, there is a decline in H_2_O_2_ levels up to 100 mM NaCl, but significant reduction in RL (100 mM: 4.5 cm) and RFW (100 mM: 51 mg/plant) were evident in comparison to V1 [RL: (100 mM: 6.0 cm), RFW: (60.8 mg/plant)] and V3 [RL: (100 mM: 6.6 cm), RFW: (71.5 mg/plant)]. These data suggested that V2 can manage H_2_O_2_ levels, but is susceptible to other NaCl induced osmotic and ionic effects [60].

It has been proposed that a reduction in RL in saline regimes is an adaptive response to maintain turgor [19], but beyond a threshold, it inhibits crop survival. Reduction in RL occurred in all the varieties with an increase in NaCl concentration with a concomitant rise in GPOD specific activity (Figure 3B). GPOD specific activity increased in all the varieties with an increase in NaCl concentration. In *T. aestivum* L. em. Thell seedlings, GPOD specific activity increased by 3 times within 20 min exposure to 150 mM and has been shown to be an outcome of ionic effect in *O. sativa* L. [17,18]. The reduction in RL in saline conditions has been shown to be associated with GPOD mediated stiffening of the cell wall [17,18]. Higher GPOD specific activity in V1 (0.43 µmole min^−1^ mg^−1^ protein) and V3 (0.42 µmole min^−1^ mg^−1^ protein) in comparison to V2 (0.35 µmole min^−1^ mg^−1^ protein) and V4 (0.11 µmole min^−1^ mg^−1^ protein) at 100 mM NaCl caused only a threshold decline in root length in association with other osmotic and ionic effects, while a severe decline was evident in V2 and V4. This could be an outcome of GPOD mediated decline H_2_O_2_ levels in V1 and V3, but lower GPOD in V4 is the cause of higher H_2_O_2_ levels and subsequent decline in hydraulic conductivity and root cell expansion [17,18,60]. In V2, GPOD is higher than V4, but lower than V1 and V3, which showed that in addition to GPOD and H_2_O_2_ other factors also play a role in root growth and development in saline conditions [60].

In glycophytes, salt tolerance is attributed to root characteristics such as maintenance of osmotic adjustment, ion homeostasis, and redox potential to perform normally in salt stress [61,62]. Roots well equipped with antioxidant defense machinery showed salt tolerance by quantitative and qualitative increase in enzymatic and non-enzymatic components, while in salt sensitive species this defense is compromised [10,11].

Steady state levels of H_2_O_2_ (0.27 to 0.29 nmole g^−1^ Fwt.) were detected in roots of all the IH varieties in control conditions (Figure 2A). However, these levels declined in V1 (1.8 times), V2 (1.2 times), and V3 (1.8 times) up to 100 mM NaCl in comparison to controls, followed by a sharp increase (0.57 to 0.62 nmole g^−1^ Fwt.) at 150 mM NaCl. In comparison to other varieties, V4 roots had a consistent rise in H_2_O_2_ levels that peaked (0.78 nmole g^−1^ Fwt.) at 150 mM NaCl. Similar increases in H_2_O_2_ levels were observed in 150 mM NaCl treated roots of *O. sativa* L. and *Lemna minor* [39,63].

H_2_O_2_ acts as a signaling molecule at low concentrations, but at elevated levels creates impaired root growth and causes programmed cell death and necrosis [10,26]. This finding is evident in current work, where higher levels of H_2_O_2_ coincided with reduced RL in V4 (Figure 1B and Figure 2A). Similarly, the highest levels of H_2_O_2_ severely reduced the RL and RFW in all the varieties at 150 mM NaCl (Figure 1B,C). Lower levels of H_2_O_2_ in V1, V2, and V3 up to 100 mM NaCl showed that these varieties are better protected from the ill effects of H_2_O_2_. A failure to check H_2_O_2_ levels lead to production of highly reactive ^•^OH through Fenton or Haber-Weiss reaction in the presence of free Cu^+^ or Fe^2+^ [21]. Hydroxyl radicals cause lipid peroxidation and perpetuate a chain reaction that generates a variety of reactive electrophiles, which in turn damage multiple cell organelles, membranes, proteins, and the DNA of a cell [21].

Lipid peroxidation is measured as MDA content, which is a marker of oxidative damage in a cell [27]. The rate of lipid peroxidation was higher (100 mM: 6 nmole g^−1^ Fwt.) in the roots of V4 at all the tested NaCl concentrations and coincided with higher levels (100 mM: 0.54 nmole g^−1^ Fwt.) of H_2_O_2_ (Figure 2A,B). However, the MDA content declined in V1 (3.1 nmole g^−1^ Fwt.) and V3 (3.2 nmole g^−1^ Fwt.) at 100 mM NaCl and coincided with lower levels of H_2_O_2_ (0.16 nmole g^−1^ Fwt.). In V2, the levels of MDA increased (3.7 nmole g^−1^ Fwt.) at 100 mM NaCl while the levels of H_2_O_2_ (0.22 nmole g^−1^ Fwt.) were low, which indicates the existence of enzymatic pathways of lipid peroxidation, independent of H_2_O_2_ levels.

Lipid peroxidation is also mediated by salt induced lipoxygenase activity [29]. Recently, lipoxygenase induced formation of ^1^O_2_ is detected in osmotically stressed roots of *Arabidopsis thaliana* [31]. These ^1^O_2_ molecules are the outcomes of the Russell reaction, occurring between the fatty acid hydroperoxide products of lipoxygenase [64]. Singlet oxygen causes lipid peroxidation and root growth inhibition [31,65]. It is possible that lipid peroxidation in V2 is mediated by H_2_O_2_ and light independent lipoxygenase-based Russell type reaction. It is well documented that salt sensitive crop varieties had higher MDA and lipoxygenase activity in comparison to tolerant varieties [66]. Roots in varieties V1 and V3 showed salt tolerance up to 100 mM NaCl by minimizing the rate of lipid peroxidation and H_2_O_2_ generation. However, this capability was abolished at 150 mM NaCl. Salt sensitive varieties V4 and V2 failed to manage the lower levels of lipid peroxidation and H_2_O_2_.

H_2_O_2_ is mainly generated by the action of SODs (Cu-Zn-SOD, Fe-SOD, and Mn-SOD), located in multiple cellular compartments such as cytoplasm, mitochondria, peroxisomes, apoplast, and chloroplast [11,21]. NaCl induced SOD activity increased in all the IH varieties up to 100 mM NaCl and thereafter declined (Figure 2C). The highest SOD activity was observed in V3 (100%), V1 (70%), and V2 (78%) and the least in V4 (51%) in comparison to controls at 100 mM NaCl. Higher SOD activity protects roots from O^•−^ Root cells produce O^•−^ in saline conditions through membrane bound NADPH oxidase [11,24]. Moreover, salt stress also accelerates the rate of respiration and causes leakage of electrons from the mitochondrial electron transport chain and generates O^•−^ [60,67,68].

SODs transform O^•−^ into less toxic H_2_O_2_ [69]. Transgenic plants with higher SOD activity showed tolerance against salinity and improvement in the anatomical structures of roots in *Populus deltoides* and *L. esculentum* L. [69,70,71]. *P. deltoides* transformed with an *MnSOD gene* (*TaMnSOD*) from *Tamarix androssowii* showed enhanced (1.3- to 4-fold) SOD activity and lower MDA content compared to wild type plants in salt stress [71]. Similarly, *L. esculentum* L. (Belyi Naliv) plants transformed with *Fe-Sod* gene from *Arabidopsis thaliana* showed ordered cytoskeleton organization in root cells in comparison to wild types at 99 mM NaCl [69]. We expect higher SOD activities in V1, V2, and V3 play an important protection role against O^•−^. However, in V4, lower SOD activities in comparison to other varieties are not sufficient to protect the roots from salinity induced O^•−^. In fact, V4 is exposed to O^•−^_,_ H_2_O_2_, and probably higher H_2_O_2_ generated ^•^OH. SOD activity declined in all the varieties at 150 mM NaCl and coincided with the higher levels of H_2_O_2_ (Figure 2A,C). Both Cu-Zn-SOD and Fe-SOD are sensitive to higher levels of H_2_O_2_ [72]. In addition, osmotic and ionic effects of NaCl directly affected the integrity of SOD protein [20].

There is a decline in H_2_O_2_ levels with a corresponding increase in SOD activity in roots of V1, V2, and V3 up to 100 mM NaCl that showed operational antioxidant defense machinery in the roots of IH varieties except V4. Salt tolerant plants maintain physiological levels of H_2_O_2_ by deploying multiple enzymatic (SOD, CAT, GPOD, APOD, and GR) and non-enzymatic (ascorbate and glutathione) components of an antioxidant defense system [60]. CAT is a monofunctional enzyme and converts two molecules of H_2_O_2_ into one molecule of water and molecular oxygen each [73]. CAT removes the bulk of the H_2_O_2_ from the system, but has a low affinity for H_2_O_2_ due to its high K_m_ value for H_2_O_2_ [74]. However, CAT does not require reducing power to remove H_2_O_2_ from the system in comparison to APOD [75]. Higher CAT activity in the roots of V1 (0.09 µmole min^−1^ mg^−1^ protein), V2 (0.08 µmole min^−1^ mg^−1^ protein), and V3 (0.09 µmole min^−1^ mg^−1^ protein) eliminated the excess of H_2_O_2_ at 100 mM NaCl (Figure 3A). However, significant decline (32%; 100 mM NaCl) in CAT specific activity in V4 at 80 mM NaCl onward caused higher accumulation of H_2_O_2_ in roots.

Recently, a bifunctional CAT is reported in the roots of *Lactuca satvia* and found to be located in mitochondria [74]. This bifunctional (catalase-peroxidase) enzyme is expected to remove H_2_O_2_ generated in the mitochondria of salt-stressed IH root cells. There is a report, which showed NaCl (80 mM) induced H_2_O_2_ formation in the mitochondria of *L. esculentum* L. root cells [10]. CAT activity declined in the roots of all the varieties at 150 mM NaCl because of salinity induced inhibition of turnover rate or synthesis of new enzymes [76].

In comparison to CAT, GPOD also removes H_2_O_2_ from the root cells through a peroxidative cycle, which requires an electron donor in the form of a phenolic compound or structural protein [36]. GPOD activity increased in roots of all the varieties with an increase in NaCl concentration (Figure 3B). However, the increase was the highest in V1 (11 times) and V3 (11 times) followed by V2 (7 times) and V4 (3 times) at 150 mM NaCl. A minimum increase in GPOD activity was observed in V4. Higher GPOD activity in association with CAT removed the bulk of H_2_O_2_ from roots in V1, V2, and V3, but poor CAT and GPOD levels in V4 compromised its antioxidant defense with a concomitant increase in root H_2_O_2_. Transgenic *Glycine max* L. plants with overexpression of *GsPRX9* from *Glycine soja* confer salinity (180 mM NaCl) tolerance in both salt sensitive and salt tolerant soybean lines. However, overexpression of this gene in salt tolerant soybean caused an increase in RFW, RL, GPOD, CAT, and SOD with lower levels of H_2_O_2_ in comparisons to controls [77]. These data are complementary to our results in the roots of IH. The varieties, V1 and V3, maintained higher GPOD, CAT, and SOD activity followed by lower H_2_O_2_ and MDA levels and minimum reduction in RL and RFW in comparison to V4 at 100 mM NaCl. GPOD is the only enzyme that consistently increased up to 150 mM NaCl salt concentration and its activity was the highest in comparison to CAT and APOD in V1, V2, and V3 (Figure 3A–C). GPOD can be used as a salt tolerance marker in the roots of IH.

APOD fine tunes the levels of H_2_O_2_ in plant cells due to its low K_m_ and high affinity for H_2_O_2_ [33]. In contrast, CAT has low affinity for H_2_O_2_ and cannot reduce its concentration to physiological levels, which means plants can suffer from oxidative stress even with higher CAT activity [33]. APOD activity increased in the roots of V1 (71%), V2 (62%), and V3 (66%) up to 100 mM NaCl while declining (22%) in V4 (Figure 3C). Enzymatic antioxidant machinery including SOD, CAT, GPOD, and APOD work in a cooperative manner in the roots of V1, V2, and V3 to manage NaCl induced H_2_O_2_. However, this system failed in the roots of V4 at all the enzymatic levels (Figure 2C and Figure 3A–C). Higher APOD activity and gene expression in roots has been corroborated with salt tolerance in *O. sativa* L., *L. esculentum* L., and *Arabidopsis thaliana* [11,42,78]. NaCl at 150–200 mM induced the expression of *OsAPx8* and activity in roots of etiolated rice seedlings [42]. Similarly, peroxisomes and mitochondria isolated from the roots of salt tolerant *L. pennellii* grown in 100 mM NaCl showed higher APOD, SOD, and CAT activity. These plants maintained lower levels of H_2_O_2_ and MDA in roots [11]. However, MDA and H_2_O_2_ levels were higher in salt sensitive *L. esculentum* L. species. APOD activity declined at 150 mM NaCl in all the varieties, which might be an outcome of higher H_2_O_2_ accumulation and non-availability of reduced ascorbate due to salt induced perturbation in cellular redox [60,79]. In the absence of reduced ascorbate, H_2_O_2_ inactivates APOD. In vitro studies showed that APOD lost its activity within 1 min when exposed to 20 mol equivalent of H_2_O_2_ [79,80].

APOD works in association with GR through the ascorbate–glutathione pathway [38]. This pathway operates at all the potential ROS production sites in a plant cell including cytoplasm, chloroplast, peroxisomes, and mitochondria [38]. APOD functioning required reduced ascorbate, which is indirectly provided by reduced glutathione by the action of GR using NADPH via an intermediate dehydroascorbate reductase reaction [37,38]. Failure of APOD and GR together and alone can cause severe oxidative damage to roots in salt stress regimes, therefore relative quantitative changes and cooperation among these enzymes determine the salt tolerance potential in plants [37,81].

GR specific activity consistently increased in V3 (6-times) and V1 (7.3-times) up to 150 mM salt concentration while it slightly declined in V2, but remained higher than controls (Figure 4). However, the decline was 2.4 times in V4 at 150 mM NaCl. The functioning of the APOD-GR cycle is futile at 150 mM NaCl in the roots of V1, V2, and V3 as higher GR activity is not complemented by higher APOD activity. In fact, severe decline in APOD activity was observed at 150 mM NaCl in all the varieties. However, similar situations exist at 80 mM NaCl in V4. This result explains higher accumulation of H_2_O_2_ in roots of V4 at 80 mM NaCl onward and in V1, V2, and V3 roots at 150 mM NaCl.

The APOD-GR cycle is fully compromised in the roots of V4 at 80 mM NaCl and 150 mM in V1, V2, and V3, which is partly responsible for the higher H_2_O_2_ accumulation in roots. Roots in V1, V2, and V3 are protected from the deleterious effects of H_2_O_2_ by a fully functional APOD-GR cycle at 100 mM NaCl, which is evident from the higher APOD and GR activities and associated lower accumulation of H_2_O_2_.

GR is an indispensable enzyme and maintains redox in multiple cellular compartments [82]. Plants with higher GR activity and gene expression showed salt resistance [82]. The *gr3* knockout mutant in rice showed 20% reduction in GR activity, higher levels of MDA, and reduced root growth at 100 mM NaCl in comparison to wild types [83]. However, complementation of *gr3* with wild type *GR3* restored the GR activity and salt tolerance. Similar to GPOD, GR activity remained higher in the roots of V1 (24 nmole min^−1^ mg^−1^ protein), V2 (19 nmole min^−1^ mg^−1^ protein), and V3 (23 nmole min^−1^ mg^−1^ protein) up to 150 mM NaCl in comparison to controls and all other enzymes (SOD, CAT, and APOD). This outcome indicates that GR confers salt tolerance in IH and can also be used as a marker for salt tolerance.

A new glutathione reductase gene (*psgr*) isolated from an Antarctic bacterium (*Psychrobacter* sp. ANT206) and cloned in *Escherichia coli* showed higher GR activity and confered salt tolerance up to 3 mM NaCl [84]. Among tested IH varieties, V1 and V3 showed improved root growth up to 100 mM NaCl in terms of better SGP, RL, RFW, and enzymatic antioxidant defense system in comparison to V4. Roots in V2 showed better oxidative stress management, but had poor RL and RFW in comparison to V1 and V3, which showed this variety may not be able to counter the other adverse effects of salinity such as ion toxicity and osmotic stress [60]. The roots in V4 are the most susceptible to salt stress. These roots displayed a maximum decline in SGP, RL, RFW, and oxidative stress indices. Based on these results, V1 and V3 can perform better up to 100 mM NaCl, V2 up to 80 mM, and V4 remained susceptible to salt stress.

## 5. Conclusions

IH showed varietal differences in salt tolerance by managing salt induced oxidative stress in roots. Varieties V1 and V3 outperforms V2 and V4 by minimum reduction in SGP, RL, and RFW, which indicates that these two varieties managed both ionic and osmotic stress and associated oxidative stress. However, V2 partly managed oxidative stress, but failed to prevent reduction in RL and RFW, associated with ionic and osmotic stress. V4 roots were compromised in saline conditions and are not suitable for growing in salt affected soils.

## Figures and Tables

**Figure 1 antioxidants-11-00244-f001:**
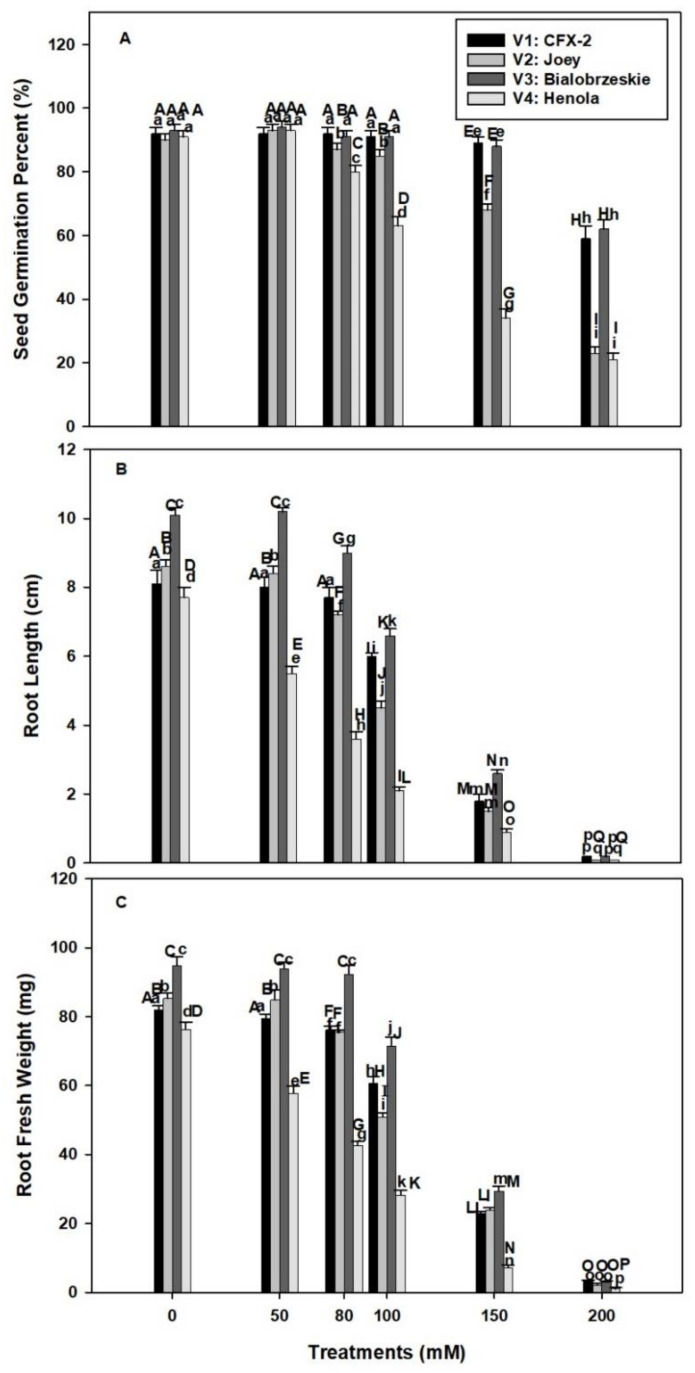
Effects of NaCl on IH SGP (**A**), RL (**B**), and RFW (**C**). Vertical bars represent SE. Different capital letters indicate significant differences among the varieties and lower case letters represent significant treatment differences. These mean values (*n* = 45) are separated using LSD at *p* < 0.05.

**Figure 2 antioxidants-11-00244-f002:**
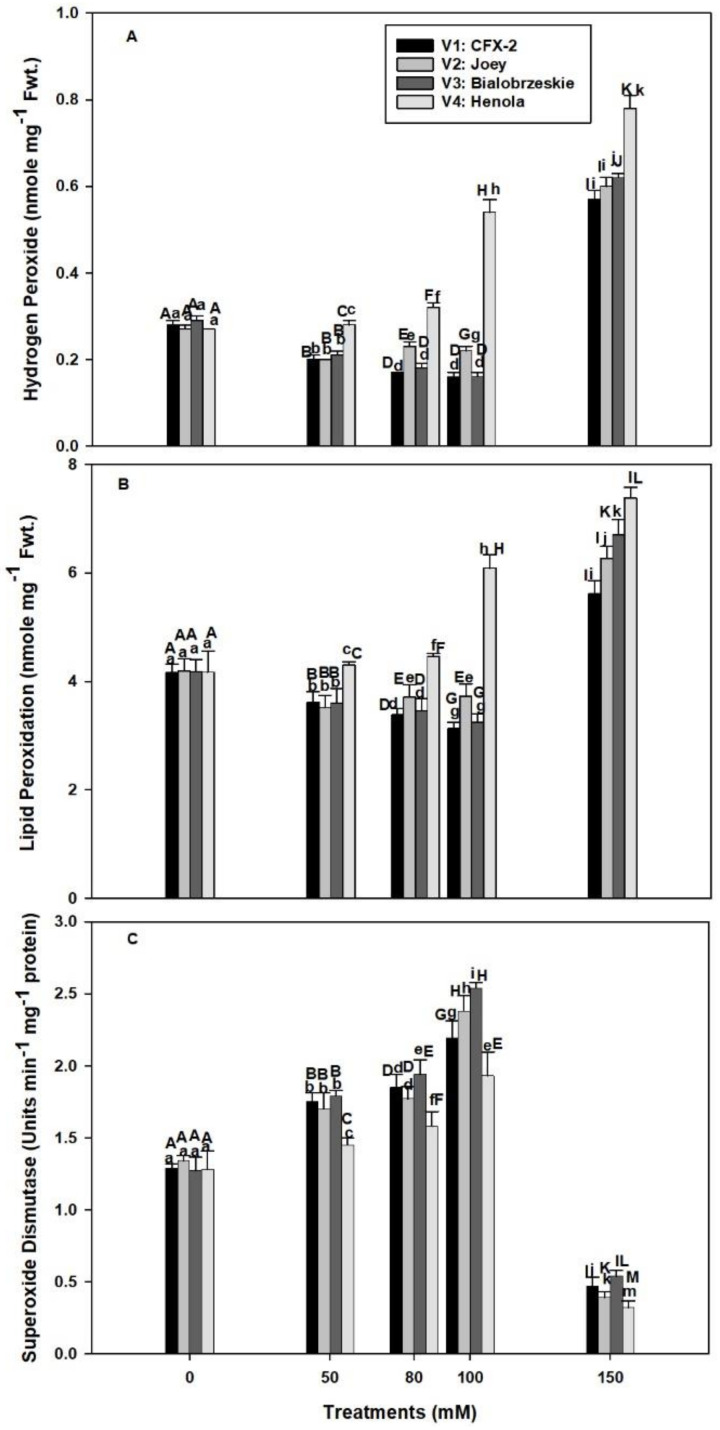
Effects of NaCl on IH root H_2_O_2_ concentration (**A**), lipid peroxidation (**B**), and SOD specific activity (**C**). Vertical bars represent SE. Different capital letters indicate significant differences among the varieties and lower case letters represent significant treatment differences.

**Figure 3 antioxidants-11-00244-f003:**
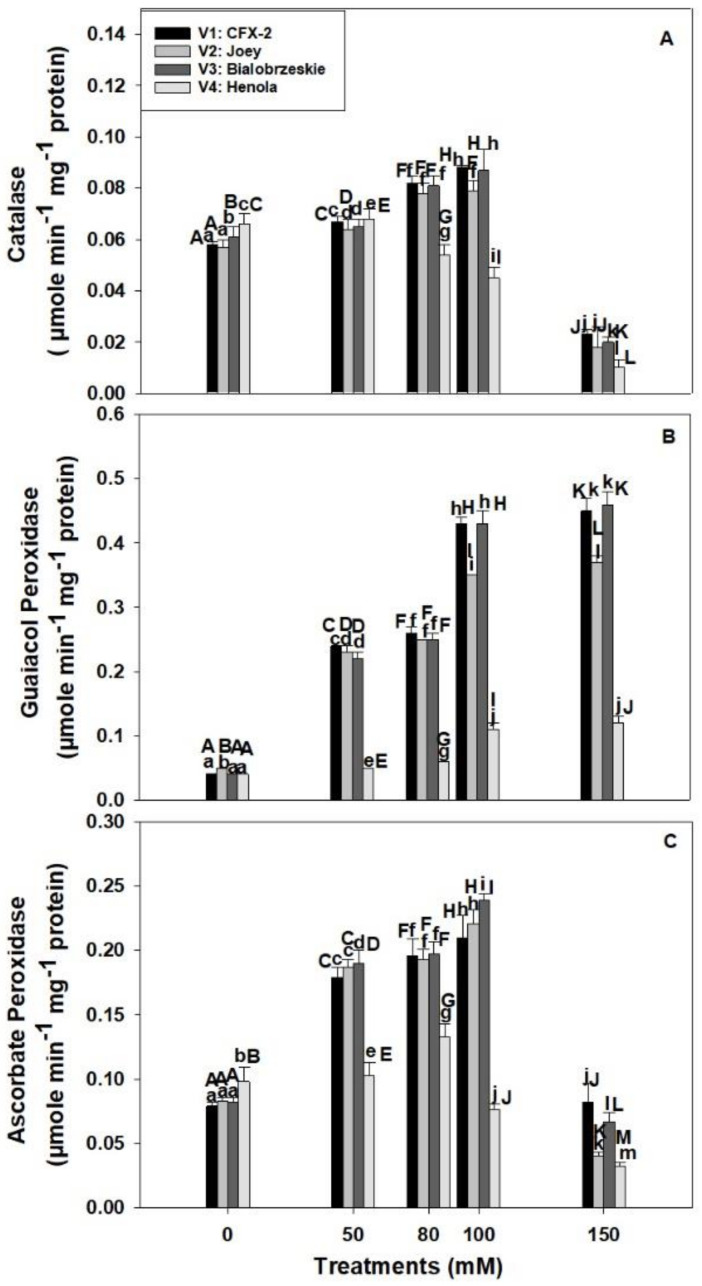
Effects of NaCl on IH root CAT specific activity (**A**), GPOD specific activity (**B**), and APOD specific activity (**C**). Vertical bars represent SE. Different capital letters indicate significant differences among the varieties and lower case letters represent significant treatment differences. These mean values (*n* = 15) are separated using LSD at *p* < 0.05.

**Figure 4 antioxidants-11-00244-f004:**
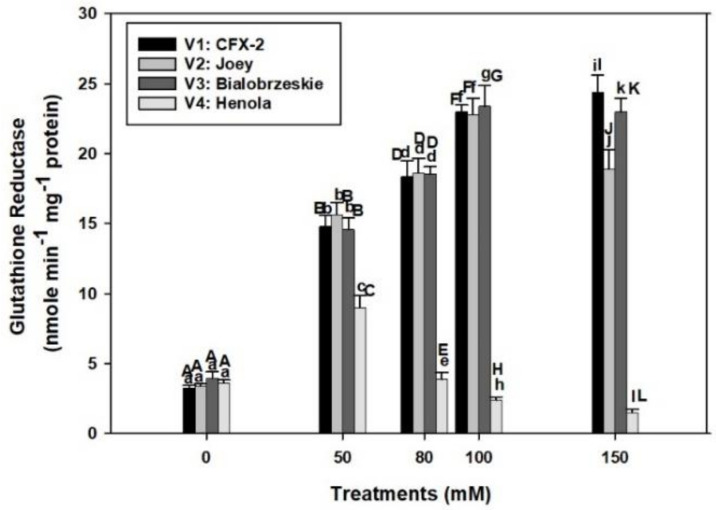
Effects of NaCl on IH root GR specific activity. Vertical bars represent SE. Different capital letters indicate significant differences among the varieties and lower case letters represents significant treatment differences. These mean values (*n* = 15) are separated using LSD at *p* < 0.05.

## Data Availability

The data presented in this study are available in the article.

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
