# Peer review of "Salinity Induced Antioxidant Defense in Roots of Industrial Hemp (IH: Cannabis sativa L.) for Fiber during Seed Germination"

_antioxidants, 2022, doi:10.3390/antiox11020244_

Round 1

Reviewer 1 Report

The review of the research article: "Salinity Induced Antioxidant Defense in Roots of Industrial Hemp (IH: Cannabis sativa L.) for Fiber During Seed Germination".

The experiment is relatively simple but well thought out. The Author shows in-depth knowledge of the subject, especialy in Discussion and selection of references. In my opinion, apart from the contribution to the studies on the plant responses to salt stress and the use of a relatively rarely analyzed plant species, the research article allows for a more general observation: such significant differences in the reaction to NaCl among the varieties and the doses used show how many simplifications we have to deal with during the analysis of numerous literature data.

The discussion is extensive and detailed. In some fragments, however, difficult to follow; therefore it would make sense to break at least the last paragraph into several parts.

I do not find any factual errors in this work, but I have some minor remarks that may help the manuscript to be more accurate and clear.

Figures would be easier to analyze if the panels were wider. Then each bar could also be wider, which would improve the legibility of the markings above the bars.

The Author analyzed the activity of four antioxidant enzymes. I do not understand why the graph showing changes in glutathione reductase activity is large and placed in a separate figure, while the activities of other enzymes are part of Figure 3 and smaller (descriptions of both figures are analogous). Is it because the CAT, GPOD and APOD reactions use a common substrate? On the other hand, in line 523 the Author recalled: "APOD works in association with GR via the ascorbate-glutathione pathway or the Foyer-Halliwell-Asada pathway [38]". Currently, the data presented in that way suggests that the graph showing GR activity is for some reason more important which is not reflected in the text.

The descriptions of the Y axis in Fig. 3 are of different sizes.­

Titles of the same "rank" should be presented uniformly. Section 3.1 is straightened, bolded and left with a blank line below, while the others (3.2, 3.3...) are italicized, with no bold or extra lines. The current layout of the Results section is messy. I wonder if it would be better to organize the results according to the division presented in the Abstract, i.e .:

3.1. Seed germination percent

3.2. Quantitative morphological observations

3.3. Oxidative stress indices

3.4. Antioxidant defense system

When mentioning the chemical symbol of the superoxide anion radical in the text, the Author should enlarge the radical's mark because it is almost invisible (compared to hydroxyl radical).

Line 106: The name of author (Akbari) should be deleted.

Line 148: There is an error in the ordering of paragraphs in the Materials and Methods section. 2.3 appears directly after 2.1.

Line 224: In the title of paragraph 3.1. Percentage of seed germination (%), I would delete (%) to avoid pairing up words and notations with the same meaning.

2022 Jan. 16th­­­­­­

Author Response

Reviewer 1:

1: The discussion is extensive and detailed. In some fragments, however, difficult to follow; therefore it would make sense to break at least the last paragraph into several parts.

Last paragraph is divided in to 5 paragraphs. Line 514-547.

2: Figures would be easier to analyze if the panels were wider. Then each bar could also be wider, which would improve the legibility of the markings above the bars.

I made these graphs using sigma plot and used the best dimensions to fit in the curve. No further increase occurred in the width of the column and bar. To further make it easier to read I also changed the font size of letters (8 to 10) and made bold in revised manuscript. Now in each graph it is easy to see significant differences. 

3: The Author analyzed the activity of four antioxidant enzymes. I do not understand why the graph showing changes in glutathione reductase activity is large and placed in a separate figure, while the activities of other enzymes are part of Figure 3 and smaller (descriptions of both figures are analogous). Is it because the CAT, GPOD and APOD reactions use a common substrate? On the other hand, in line 523 the Author recalled: "APOD works in association with GR via the ascorbate-glutathione pathway or the Foyer-Halliwell-Asada pathway [38]". Currently, the data presented in that way suggests that the graph showing GR activity is for some reason more important which is not reflected in the text.

I tried to group figures in a group of three so the last remaining figure was for GR. There is no such intention to separate the figure of GR from CAT/GPOD/APOD. In fact, GR is important for GR-APOD cycle. Almost last 5 paragraphs in discussion section is devoted to GR only.

Line number:  520-553

4.The descriptions of the Y axis in Fig. 3 are of different sizes.­

In Fig. 3, all the Y axis titles has same size and font in revised manuscript. Line 291.

5: Titles of the same "rank" should be presented uniformly. Section 3.1 is straightened, bolded and left with a blank line below, while the others (3.2, 3.3...) are italicized, with no bold or extra lines. The current layout of the Results section is messy. I wonder if it would be better to organize the results according to the division presented in the Abstract, i.e .:

3.1. Seed germination percent, 3.2. Quantitative morphological observations, 3.3. Oxidative stress indices, 3.4. Antioxidant defense system

Section 3.1 is now similar to all other headings in the result section. There is no extra spacing. All headings are same. This is a good suggestion to make 4 categories, but this will further make sub-section to classify root fresh weight and root length under quantitative morphological observations. Similarly, more sub-sections in oxidative stress indices and can make it more complicated.

Line 215- 319

  1. When mentioning the chemical symbol of the superoxide anion radical in the text, the Author should enlarge the radical's mark because it is almost invisible (compared to hydroxyl radical).

Thanks for this suggestion. It is changed and made bold OË™‾ throughout the manuscript. It was earlier like this OË™‾.

  1. Line 106: The name of author (Akbari) should be deleted.

Akbari deleted and replaced by ‘salt sensitive cultivars’ in the text. Line 106.

8: Line 148: There is an error in the ordering of paragraphs in the Materials and Methods section. 2.3 appears directly after 2.1.

Mistake is fixed in the revised manuscript. The new numbering varies from 2.1 to 2.5. Line 123-206

9: Line 224: In the title of paragraph 3.1. Percentage of seed germination (%), I would delete (%) to avoid pairing up words and notations with the same meaning.

Line number 216: % sign deleted in section 3.1.

Thanks for your critical suggestions.

Reviewer 2 Report

This research evaluated the salt-tolerance capability based on the 5 salinity treatments. Morphological and physiological traits were used to screen the IH varieties. At last, the varieties V1 and V3 showed more salt tolerance. The result provided new IH materials to plant in saline-alkali land. Yet, for all of that, this study just simply compared the morphological and physiological traits without any biostatistics analysis or genetic analysis. It's better to carry out more biostatistics analysis such as genetic component analysis or genetics analysis such as RNA-seq and so on.

Author Response

Reviewer 2:

This research evaluated the salt-tolerance capability based on the 5 salinity treatments. Morphological and physiological traits were used to screen the IH varieties. At last, the varieties V1 and V3 showed more salt tolerance. The result provided new IH materials to plant in saline-alkali land. Yet, for all of that, this study just simply compared the morphological and physiological traits without any biostatistics analysis or genetic analysis. It's better to carry out more biostatistics analysis such as genetic component analysis or genetics analysis such as RNA-seq and so on.

This is the first study in IH for fiber, which dissected the oxidative stress machinery and antioxidant defense system in roots. There is not a single published article available on IH roots, which explored the oxidative stress management in saline regimes.

This is a good suggestion, but we do not have capability, expertise, and resources to perform genetic component analysis and RNA-seq. However, we analyzed our data using accepted/published statistical design and this statistical design is used multiple times for such type of data publication in the journal ‘Antioxidant’. Our design showed statistical difference among varieties and treatments. Some of the recent examples are listed below, which are published in Antioxidant similar to our work in IH.

1: Nitric Oxide Ameliorates Plant Metal Toxicity by Increasing Antioxidant Capacity and Reducing Pb and Cd Translocation. Antioxidants 2021, 10(12), 1981. 

2.9. Statistical Analysis

The data analysis was conducted under a completely randomized design (CRD) through a 2-way factorial design with four replicates. The statistical package of R software was allocated to the analysis of variance (ANOVA). Tukey’s test was supplied for the comparison of the mean difference at the p < 0.05 probability level.

2: Strigolactones Modulate Cellular Antioxidant Defense Mechanisms to Mitigate Arsenate Toxicity in Rice Shoots. Antioxidants 2021, 10(11), 1815. 

2.8. Data Analysis

The Statistix 10 software was used to carry out a two-way analysis of variance (ANOVA) on all of the data. Arithmetical data are provided as means with standard errors (SEs). For physiological and biochemical parameters, and expression of associated genes, the least significant difference (LSD) post hoc test was carried out to identify significant variations among the treatments (p < 0.05). For analyzing expression data of D10 and D17 genes, the Student’s t-test (** p < 0.01) was conducted to identify significant variations among As0, As1 and As2 treatments.

3: Antioxidative Responses of Duckweed (Lemna minor L.) to Phenol and Rhizosphere-Associated Bacterial Strain Hafnia paralvei C32-106/3. Antioxidants 2021, 10(11), 1719 

2.10. Statistical Analysis

Each experiment was repeated independently at least three times. The number of replicates is indicated in the caption of each figure. Numerical data were analyzed using the computer program GraphPad Prism 9.1.0, San Diego, CA, USA. Each value represents the mean ± standard error (SE). The significance of difference of treatments was tested by Student’s paired t-test. The difference was considered significant at p less than 0.05 (p < 0.05).

Round 2

Reviewer 2 Report

The revised manuscript had explained my concern and looking forward to furthering research.